# Effect of Cashew Nut Consumption on Biomarkers of Copper and Zinc Status in Adolescents with Obesity: A Randomized Controlled Trial

**DOI:** 10.3390/nu17010163

**Published:** 2024-12-31

**Authors:** Thaynan dos Santos Dias, Kaluce G. de S. Almondes, Matheus A. D. Firmino, Luis Felipe N. de Oliveira, Walter B. de S. Freire, Fernando Barbosa, Maria Dinara de A. Nogueira, Liliane V. Pires, Vicente Martínez-Vizcaíno, Arthur E. Mesas, Luis A. Moreno, Carla S. C. Maia

**Affiliations:** 1Health Sciences Center, Universidade Estadual do Ceará, Fortaleza 60714-903, Brazil; thaynan.dias@aluno.uece.br (T.d.S.D.); felipee.oliveira@aluno.uece.br (L.F.N.d.O.); carla.maia@uece.br (C.S.C.M.); 2Micronutrients and Non Communicable Disease Research Group, Universidade Estadual do Ceará, Fortaleza 60714-903, Brazil; kaluce@gmail.com (K.G.d.S.A.); matheus.firmino@usp.br (M.A.D.F.); dinara.araujo@aluno.uece.br (M.D.d.A.N.); 3Faculty of Medicine, Universidade de São Paulo, Ribeirão Preto 14040-900, Brazil; 4Integrated Healthcare Center—Nami, Universidade de Fortaleza, Fortaleza 60811-905, Brazil; walterbreno@unifor.br; 5School of Pharmaceutical Sciences, Universidade de São Paulo, Ribeirão Preto 14040-900, Brazil; fbarbosa@fcfrp.usp.br; 6Postgraduate Program in Nutrition Sciences, Universidade Federal de Sergipe, São Cristóvão 49100-000, Brazil; lvianapires@gmail.com; 7Health and Social Research Center, Universidad de Castilla-La Mancha, 13071 Cuenca, Spain; vicente.martinez@uclm.es; 8Faculty of Health Sciences, Universidad Autónoma de Chile, Talca 3465548, Chile; 9Growth, Exercise, Nutrition and Development Research Group, School of Health Science, Universidad de Zaragoza, 50009 Zaragoza, Spain; lmoreno@unizar.es

**Keywords:** anacardium, nuts, minerals, pediatric obesity, superoxide dismutase, micronutrients, regional food

## Abstract

Background: Children and adolescents with obesity have altered serum copper (Cu) and zinc (Zn) levels, which are associated with oxidative stress, inflammation, and health outcomes. The inclusion of cashew nuts in an adequate diet may provide health benefits and help improve the mineral status of individuals with obesity. Objective: To evaluate the effects of cashew nut consumption on biomarkers of Cu and Zn status in adolescents with obesity. Methods: This was a randomized controlled trial conducted in adolescents. The participants were divided into a control group (CON) and a cashew nut group (CASN). The CASN group received 30 g/day of roasted cashew nuts for 12 weeks, and both groups received nutritional counseling during the study. Anthropometric, dietary, and biochemical parameters (Zn, Cu, and superoxide dismutase (SOD)) were assessed at the baseline and at the end of the study. Association tests, within-group and between-group mean comparisons, and analyses of variation between study periods (Δ T12-T0) were performed. Results: The sample consisted of 81 adolescents who completed the intervention period, resulting in 54 in the CASN group and 27 in the CON group. After the intervention, the CASN group presented a decrease in plasma Cu (*p* = 0.004) and an increase in SOD (*p* = 0.030). Both groups showed an increase in plasma Zn (*p* < 0.050) and a decrease in the Cu/Zn ratio (*p* < 0.001). CASN had a negative effect on the Cu concentration, which was significantly different from that of CON (*p* = 0.004). Conclusions: The consumption of cashew nuts for 12 weeks reduced plasma Cu levels in adolescents with obesity. Nutritional counseling may have contributed to the increase in plasma Zn levels in all the study participants.

## 1. Introduction

Obesity is a complex multifactorial disease characterized by the expansion of adipose tissue with an inflammatory phenotype [1]. In children and adolescents, this disease has become a serious global public health problem because of its alarming progression and serious short- and long-term health consequences [2]. Research from different countries shows that obesity is associated with an altered pattern of copper (Cu) and zinc (Zn) homeostasis during childhood and adolescence [3,4,5,6]. Although some studies have shown conflicting results [7,8], meta-analyses have confirmed alterations in serum or plasma concentrations in this age group, concluding that Cu levels are increased while Zn levels are decreased [9,10].

In addition to obesity, studies show that this altered pattern of Cu and Zn is found in other diseases, such as diabetes mellitus [11,12], metabolic syndrome [13], and nonalcoholic fatty liver disease [14]. In particular, serum Cu levels ≥ 133.9 μg/dL in overweight/obese individuals have been associated with an increased risk of all-cause mortality [15].

These micronutrients are essential and have critical functions in biological processes. Cu plays a catalytic role in cuproenzymes, performing redox cycles necessary for cellular respiration, free radical detoxification, and neurotransmitter biosynthesis [16]. Zn is a component of more than 300 enzymes [10] and plays important roles in transcription factors, DNA-binding proteins, and cellular regulation through kinases, phosphatases, and transport channels [17].

Cu and Zn are important components in the control of redox balance and are crucial for the management of oxidative stress and the inflammatory process. The enzyme superoxide dismutase (SOD), which relies on these minerals, facilitates the conversion of superoxide radicals into hydrogen peroxide and helps maintain cellular homeostasis [18]. Both minerals also compete for intestinal absorption, and this competition can influence the availability of minerals in the body, especially in situations of unbalanced intake or deficiency [19]. Alterations in the homeostasis of these minerals may promote a more oxidative and proinflammatory state, influencing the pathogenesis of obesity and the development of clinical complications [9,10].

Diet is the primary way to obtain these nutrients and influence the status of these compounds in the body [20]. A healthy diet consisting of foods rich in micronutrients and bioactive compounds, such as nuts, can help maintain adequate vitamin and mineral status [21] and contribute to the control of inflammation and oxidative stress [22,23]. The interaction of these factors appears to be well established in the literature regarding the consumption of certain nuts, such as Brazil nuts [20,21]. However, research specifically focused on cashew nuts remains limited.

Cashew nuts are one of the most important nuts grown in northeastern Brazil. They are rich in unsaturated fatty acids, dietary fiber, vitamins, minerals, and bioactive compounds [24,25]. Due to their nutritional properties, cashew nuts have been studied for their ability to control chronic and acute inflammatory and oxidative processes. However, there is a gap in the scientific literature regarding the effects of cashew nut consumption on Cu and Zn status. Based on promising results from experimental research [26,27,28,29], we hypothesized that their consumption in combination with a healthy diet could produce similar results in a clinical setting, which in turn might contribute to the control and prevention of chronic diseases [30,31]. However, evidence concerning the effects of cashew nut consumption on the concentration of these essential minerals is still limited.

Therefore, the aim of this study was to evaluate the effects of cashew nut consumption on biomarkers of Cu and Zn status in adolescents with obesity.

## 2. Methods

### 2.1. Study Design and Location

The study is a randomized controlled trial conducted among students diagnosed with obesity in schools in Fortaleza, Ceará, Brazil. This project refers to Phase 3 of the Study of Nutrition and Health of Children and Adolescents in Fortaleza (ENSCA-FOR), which consists of three phases: (1) a cross-sectional screening study; (2) a pilot study; and (3) a randomized controlled trial (RCT). Data collection was carried out in public schools in Fortaleza, Ceará, Brazil. Phase 3 began in February 2022 and ended in December of the same year.

### 2.2. Sample Size and Participants

The selection of schools and the sample size for Phase 1 of the study are described in detail in Ricarte et al., 2024 [32]. The total number of Phase 1 participants (1483) was used to estimate the sample size for the RCT, considering a prevalence of obesity of 7.4% among adolescents in the Northeast region of Brazil [33]. A minimum of 110 participants were enrolled in the RCT.

All students screened in Phase 1 who met the inclusion criteria (377 adolescents) were invited to participate in the study. Of these, 260 students agreed to participate. They were randomly allocated to a cashew nut intervention group (CASN) and a control group (CON) under a simple 1:1 allocation ratio.

The inclusion criteria were adolescents aged 10 to 14 years, of both sexes, with body mass index (BMI) for age Z-scores greater than or equal to 2, according to the World Health Organization (WHO) curves [34]. The exclusion criteria were allergy or intolerance to cashew nuts, use of multivitamin supplements, medical treatment for a preexisting condition, pregnancy, physical or cognitive disability that could interfere with adherence to all study phases, and being under the care of a healthcare professional for weight loss.

### 2.3. Intervention

The intervention lasted 12 weeks equally divided into 4 sessions with assessments (at T0 [baseline], T4, T8, and T12), in which face-to-face meetings were conducted. The participants did not receive individualized meal plans but were encouraged to follow the nutritional advice provided at the meetings. Dietary counseling was conducted in schools with participants from both groups by nutritionists previously trained for the study.

At T0, the first meeting of the study, the following data were collected: sociodemographic variables, anthropometry, body composition, dietary intake, and biochemical variables. All participants were given a recipe book containing healthy preparations with regional ingredients from the state of Ceará, Brazil, developed by the Micronutrients and Chronic Diseases Study Group (Appendix A).

At T4 and T8, corresponding to the second and third meetings, respectively, participants’ dietary intake data were collected using 24-h dietary recalls (24HR). The researchers also conducted group nutritional guidance activities based on the “Guia Alimentar para a População Brasileira” (Dietary Guidelines for the Brazilian population) [35] and the NOVA Food Classification, according to the degree of processing [36]. Details regarding the nutritional guidance activities are available in Appendix A.

At T12, the last day of the study, the complete assessment of the students (anthropometric measurements, body composition, dietary intake, and biochemical tests) was repeated.

### 2.4. Roasted Cashew Nut

In addition, the CASN group received 30 g/day of roasted, broken, and unsalted cashew nuts (*Anacardium occidentale* L.). During the 12-week period, participants were given individually packaged servings of cashew nuts at each face-to-face meeting in amounts sufficient to consume one package per day until the next meeting date. The participants were instructed to consume one serving of cashew nuts daily at any time of day and were advised not to consume any other type of nut during the study.

Cashew nuts were obtained from a food company in Eusébio, Ceará, Brazil. The raw material was subjected to a temperature of 125 °C for 2 min and packaged in laminated containers without commercial branding. The nutritional composition of the cashew nuts provided in the study is presented in Table 1.

The analysis of macronutrients was performed according to the methods of the Adolfo Lutz Institute [37] and the Association of Official Analytical Chemists [38]. The samples were subjected to a microwave digestion process via Start D-Milestone^®^ (Milestone Srl, Sorisole, Italy), and the resulting contents were diluted in ultrapure water for reading by Inductively Coupled Plasma Optical Emission Spectroscopy—ICP–OES (PerkinElmer, Shelton, CT, USA).

To monitor adherence to cashew nut consumption, a trained team conducted weekly telephone checks with the parents or guardians and face-to-face checks with the students using 24HR during the study meetings. A minimum adherence of 80% of consumption during the study period was considered valid.

### 2.5. Ethical Aspects

This study was registered in the Brazilian Registry of Clinical Trials (ReBEC) under number RBR-7nfy2z3 and was approved by the Research Ethics Committee of the State University of Ceará under number 13338419.6.0000.5534 (CAAE). The research was conducted with the consent of the parents or legal guardians through a signed free and informed consent form, and with the consent of the adolescents through a signed informed consent form.

### 2.6. Study Variables

Sociodemographic data were obtained through interviews using a form developed by the researchers. The questions covered family and household characteristics. Anthropometric and body composition, dietary intake and biochemical marker levels were assessed at T0 and T12.

Dietary intake:

Dietary intake was assessed via the 24HR during the study meetings. The interviews followed the multiple-pass method [39]. The weights of the foods consumed were expressed in terms of energy, macronutrients, and micronutrients.

Nutritional information was based on the Brazilian Food Composition Table [40], the “Pesquisa de Orçamentos Familiares” Table (2008–2009) [41], and the United States Department of Agriculture Food Search Table [42], in that order if the information was not available in the previous reference. The consumption data were tabulated using NutWin^®^ software, version 1.5.2.11, developed in 2002 by the Federal University of São Paulo, Brazil.

Anthropometric and body composition assessment:

The participants underwent the following anthropometric measurements: weight and height. Weight was measured using a portable electronic scale (Omron^®^, OMRON Co., Kyoto, Japan) with a capacity of up to 150 kg and an accuracy of 100 g, with participants wearing light clothing and no shoes. Height was measured in a standing position using a portable stadiometer (Sanny^®^, AMERICAN MEDICAL DO BRASIL LTDA, São Bernardo do Campo, Brazil), with a range of 200 cm and a variation of 0.1 cm. These values were used to calculate BMI and to assess nutritional status using BMI for age, which was classified according to the Z-score using WHO curves [34].

For body composition analysis, participants underwent bioelectrical impedance analysis using a tetrapolar device (Biodynamics^®^ model 450, 800 μA, 50 kHz; Biodynamics Corporation at United Estates, Seattle, WA, USA). The following conditions were included in the testing protocol: no pacemaker, water and food fasting (8–12 h), no smoking for at least two hours prior to testing, empty bladder, and no exercise for at least 12 h prior to testing.

Biochemical markers:

Venous blood collection was performed after an 8–12 h fast by a nursing technician. Blood was collected using disposable plastic syringes and stainless steel needles and then transferred to polyethylene vacuum blood collection tubes containing sodium citrate anticoagulant and ethylenediamine tetraacetic acid anticoagulant.

Blood samples were centrifuged at 3500 rpm for 15 min at 4 °C to obtain plasma. After plasma separation, the erythrocytes were washed three times with 5 mL of mineral-free saline solution (0.9%), slowly homogenized each time and centrifuged at 4 °C for 15 min at 4500 rpm. The samples were then collected in polypropylene microtubes and stored in a temperature-controlled freezer for later analysis.

To minimize material contamination by minerals, all materials used in blood processing were either purchased mineral-free or subjected to a 20% nitric acid bath for at least 24 h followed by at least 10 consecutive rinses with deionized water.

Cu and Zn were analyzed in plasma samples by inductively coupled plasma–mass spectrometry with a reaction cell (DRC-ICP-MS ELAN DRCII, Perkin Elmer, Sciex, Norwalk, CT, USA) operated with high-purity argon (99.999%) (Praxair, Pinhais, Brazil). Samples were diluted 1:50 in 15 mL polypropylene Falcon^®^ tubes (Corning, Glandela, CA, USA) with a solution containing 0.01% (*v*/*v*) Triton X-100 (Triton X-100 Merck KGaA, Darmstadt, German), 0.5% (*v*/*v*) HNO_3_, and 10 µg/L^−1^ Rh as an internal standard. Calibration standards were prepared at concentrations ranging from 0 to 50 µg/L in the same diluent. Values were expressed in µg/dL, and the Cu/Zn ratio was calculated from the Cu (µg/dL) and Zn (µg/dL) values.

Quality control of the analyses was ensured by analysis of reference materials from the Quebec National Public Health Institute, Canada. Blood reference materials were analyzed for data validation (EQAS, INSP-L’Institut National de Santé Publique du Quebec, Quebec, QC, Canada), QMB-B-Q 1515, and QM-B-Q-Q1720.

The nutritional status of the participants in terms of plasma Cu and Zn was classified into categories. Cu was considered normal (or low) when values were ≤140 µg/dL for males and ≤155 µg/dL for females. Zn was categorized as normal when values were >70 µg/dL, for both sexes [43].

Superoxide dismutase (SOD) enzymatic activity was measured in erythrocyte samples using the Ransod commercial kit RANDOX^®^ (Crumlin, County Antrim, BT29 4QY, UK) according to the manufacturer’s method. The results were read on the Labmax 240 Premium biochemical analyzer from LABTEST^®^ (Lagoa Santa, Brazil). SOD values were corrected for the hemoglobin concentration of each sample, which was evaluated by the cyanmethemoglobin method using a LABTEST^®^ kit and read on a spectrophotometer at a wavelength of 540 nm. The values are expressed as U/gHb, and the normal range was adopted according to the test manufacturer: 1102–1601 U/gHb.

### 2.7. Statistical Analysis

The sociodemographic characteristics of the groups were assessed using T0 (baseline) values. The associations between categorical variables sociodemographic and the status of Cu and Zn were analyzed using Pearson’s chi-square test. The results are presented as absolute and relative (%) frequencies.

All the quantitative continuous variables were examined with the Kolmogorov–Smirnov test to assess the nature of the distributions and Levene’s test to determine homogeneity. The data are presented as the means and standard deviations (SDs).

The variables were subjected to the intent-to-treat method, where missing data were imputed using the complete database to obtain postintervention (T12) values from the baseline values of the participants. The SOD variable was not imputed because, due to resource limitations, biochemical tests were analyzed only for participants who both started and completed the study. The Little’s MCAR test was conducted, confirming the appropriateness of this method (Little’s MCAR test: chi-square = 87.241, DF = 94, Sig. = 0.676).

The within-group, T0 and time and group interaction differences were compared using the parametric test of mixed-design ANOVA for repeated measures with Bonferroni post hoc correction. Variables that were not normally distributed were transformed using natural logarithms.

To compare the change following the intervention between the two groups, new variables were created by subtracting the baseline values (T0) from the values at T12 (Δ T12-T0). The resulting differences between the CASN group and the CON group were tested via Student’s *t*-test for independent samples or the Mann–Whitney test, according to the data distribution.

Statistical tests were performed using the Statistical Package for the Social Sciences (SPSS Inc., Chicago, IL, USA), version 20.0 [44]. The significance level was set at *p* < 0.05.

## 3. Results

Of the 260 randomized adolescents, 143 started the study: 78 in the CASN group and 65 in the CON group. The mean age of the CASN group was 12.45 (SD 1.39) years, and that of the CON group was 12.29 (SD 1.26) years, with no significant differences between the groups. The populations were similar in terms of most of the sociodemographic characteristics assessed (Table 2).

Among these 143 participants, 81 completed the whole intervention and met the criteria for this study, corresponding to a final number of 54 (70.13%) adolescents in the CASN group and 27 (41.54%) in the CON group. One participant was excluded at week four because of ongoing medication use for a preexisting disease that was not initially reported at screening. The number of dropouts was influenced by various factors, including the absence of adolescents from school on the day of data collection, and the fear of having blood samples taken, as reported by some adolescents. More details on participant follow-up are shown in Figure 1.

During the study, all 54 adolescents had confirmed adherence to the intervention (>80%) through weekly telephone follow-ups with parents or guardians, achieving self-reported cashew consumption adherence. No adverse reactions to food consumption were reported.

Anthropometric, dietary, and biochemical variables were analyzed under the intent-to-treat principles. Both groups showed a slight increase in height and a reduction in BMI at 12 weeks. A slight change in these variables was expected due to the growth phase. There were no changes in body composition or dietary characteristics in response to the intervention (Table 3).

In the between-group analysis, the groups were not homogeneous in terms of the plasma Zn concentration or the Cu/Zn ratio. Plasma Zn concentration was significantly higher in the CON group compared to the CASN group at T0 (*p* = 0.010), while the Cu/Zn ratio was significantly lower (*p* = 0.013). With respect to the time-by-group interaction, the plasma Cu concentration was also lower in the CASN group (*p* = 0.004). In terms of within-group changes, after the intervention period, the plasma Zn concentration increased, and the Cu/Zn ratio decreased in both groups. Cu decreased (*p* = 0.007), and SOD (*p* = 0.030) increased only in CASN (Table 4).

When the classification of the Cu and Zn levels was considered, the results were consistent with the previously mentioned data (Figure 2). At T12, the prevalence of adolescents with elevated Cu levels was lower in the CASN group than in the CON group (*p* = 0.034). Zn levels were greater in the CON group. With respect to SOD, all participants were above the reference range at T0 and T12, according to the established reference values.

The variation (Δ T12-T0) in the biochemical markers is shown in Figure 3. CASN had a negative effect on the plasma Cu concentration (−6.34 µg/dL), which was significantly different from that of CON (+3.61 µg/dL) (*p* = 0.004).

## 4. Discussion

The consumption of cashew nuts promoted a reduction in plasma Cu levels in the CASN group, even without altering the total intake of this micronutrient, and plasma Zn may have changed in response to nutritional counseling. To our knowledge, this is the first study to evaluate Cu and Zn in response to cashew nut consumption in adolescents with obesity.

Cashew nuts, similar to other nuts, are rich in nutrients and bioactive compounds that exert beneficial health effects [45,46]. Given their composition, the consumption of cashew nuts, in combination with a healthy diet, such as other nuts, could help control the inflammation and oxidative stress [23,47] associated with obesity in our study population. This hypothesis was partially confirmed in our study because SOD levels increased in the group that consumed cashews for 12 weeks (CASN group) but not in the control group (CON group).

Few clinical trials have evaluated the individual effects of cashew nuts on oxidative stress and/or inflammatory markers [30,31,48], and no studies have assessed plasma Cu and Zn biomarkers. Therefore, our results cannot be compared with the current scientific literature to refute or confirm the possible effects of cashew nuts on these physiological imbalances.

Damavandi et al. [30] reported that the consumption of cashew nuts for eight weeks did not change the total antioxidant capacity or the concentrations of paraoxonase 1 and high-sensitivity *C*-reactive protein in individuals with type 2 diabetes. Davis et al. [31] reported no changes in plasma antioxidant capacity, total serum antioxidant status, or glutathione redox status in individuals with metabolic syndrome after eight weeks of cashew nut consumption. Baer and Novotny [48] reported no clinically relevant results for inflammatory markers after four weeks of cashew nut consumption in healthy individuals.

Unlike the abovementioned studies, our study proposed evaluating the effects of cashew nut consumption for a longer period of time (12 weeks) on markers of Cu and Zn. These minerals, when altered in the bloodstream, may indicate the presence of oxidative stress and systemic inflammation in children and adolescents [9,10,49].

Our results suggest that cashew nuts may modulate these factors (oxidative stress and inflammation) through Cu-related pathways by unknown mechanisms, regardless of weight loss or changes in body composition. This finding is relevant because altered plasma Cu levels are associated with the development of various chronic diseases and mortality from different causes [3,11,12,13,14,15].

The plasma Cu levels of the adolescents who consumed cashew nuts presented a negative variation in plasma Cu levels after the intervention. Importantly, cashew nuts are rich in Cu, but their consumption for 12 weeks in adolescents with obesity promoted a reduction in plasma values.

The mechanisms underlying our findings are not fully understood. However, the major circulating Cu transporter, ceruloplasmin, is elevated in the serum of individuals with obesity and is associated with inflammation [50,51]. In response to inflammation, ceruloplasmin may contribute to increased nitric oxide synthase activity and, consequently, increased nitric oxide production [52].

Under inflammatory conditions, unbound Cu may also be increased, and the mineral in its free form may act as a significant pro-oxidant factor [9,51,53]. Cu can catalyze the formation of hydroxyl radicals from hydrogen peroxide through Haber-Weiss and Fenton reactions [54]. These highly reactive radicals produce more reactive oxygen species, causing damage to cell structures.

An imbalance in redox balance can lead to oxidative stress, which increases the activation of transcription factors and signaling pathways that positively regulate proinflammatory cytokines and chemokines. Clinically, elevated Cu is associated with oxidative stress and inflammation [9]. By reducing plasma Cu, we believe that there was a decrease in reactive species production mediated by excess Cu, which may have positively influenced adolescents’ health.

The CON group presented a slight reduction in dietary Zn intake, possibly due to changes in food choices following nutritional guidance such as reducing the consumption of certain ultra-processed foods, which are often fortified with micronutrients, including zinc. On the other hand, both the CON and CASN groups presented relatively high plasma Zn concentrations and reduced Cu/Zn ratios after 12 weeks. A dietary pattern based on increased fruit and vegetable consumption and reduced consumption of processed and ultra-processed food is associated with lower levels of inflammatory biomarkers and increased levels of micronutrients [55]. In this sense, it is possible that nutritional guidance promoted the changes observed in both groups.

Cu and Zn concentrations can be modulated in response to inflammation in the body [56]. Cu is positively regulated in the bloodstream as a positive acute-phase reactant and is bound mainly to ceruloplasmin. In contrast, Zn is mainly transported by albumin and tends to decrease in response to inflammation [57]. This inverse behavior with respect to inflammatory cytokines may contribute to antagonistic changes in trace elements [10]. Thus, improving the inflammatory state may be directly related to a decrease in Cu and a concomitant increase in Zn.

In addition to changes in mineral profiles, the CASN group presented increased SOD activity. SOD1, the isoform present in erythrocytes, is an antioxidant enzyme that contains Cu and Zn, and the presence of these minerals allows the enzyme to function properly. Oxidative stress and inflammation are factors that influence increased SOD activity [56]. As observed in our study, both groups presented SOD levels above the reference range at T0 and T12.

The long-term accumulation of inflamed and dysfunctional adipocytes resulting from obesity may also lead to decreased antioxidant enzymes due to overload. Thus, SOD levels may not be maintained at levels sufficient to control superoxide production [58,59]. We believe that a possible improvement in the inflammatory state, as evidenced by reduced plasma Cu and improved Zn status, may have contributed to increased SOD activity only in the CASN group.

Zn is essential for several antioxidant and anti-inflammatory biological processes and is considered one of the most important regulators. Zn competes with toxic and redox-active metals, including Cu, to suppress the ability of these components to generate reactive species and cause cellular damage [60]. It also binds to protein thiol groups, protecting them from oxidation. Additionally, various antioxidant and proinflammatory pathways that control the body’s redox state are indirectly regulated by this mineral [61].

The improved Cu/Zn ratio supports our hypothesis regarding the mineral response to the intervention. The ratio of these metals is recognized as a potential biomarker of nutritional and inflammatory status in children and adolescents with chronic diseases [62]. The ratios of these trace elements are associated mainly with the inflammatory state rather than with nutritional factors [62]. Therefore, the biochemical values did not differ significantly between the two groups in terms of dietary nutrient concentrations.

Although this study has made a significant contribution, some limitations must be considered. First, there were challenges with study adherence, which seems to be characteristic of clinical trials and the administration of food to a single group and the population studied. Second, due to research resource limitations, it was not possible to assess other direct markers associated with Cu and Zn status.

This study has strengths that increase the robustness and relevance of the findings. The RCT design ensures internal validity, minimizes selection bias and allows the evaluation of cashew nut effects. Our findings are novel and provide insights for new research on the phytochemical properties of cashew nut metabolic pathways associated with inflammation and oxidative stress regulated by Cu and Zn.

We suggest that the consumption of cashew nuts, in combination with a healthy diet, may help regulate essential metals for antioxidant control and anti-inflammatory responses in obese individuals. Further studies are needed to explain the underlying mechanisms of our findings and to identify the components in cashew nuts that can modify the serum profile of these minerals.

## 5. Conclusions

The consumption of cashew nuts reduced plasma Cu levels in adolescents with obesity. On the other hand, nutritional guidance activities may influence increased plasma Zn levels and reduced Cu/Zn ratios in adolescents with obesity. Notably, similar results were observed in both the CASN and CON groups with respect to body composition and Cu and Zn intake. Therefore, further studies are needed to investigate the potential mechanisms underlying changes in Cu and Zn levels following cashew nut consumption, with particular focus on their roles in regulating inflammation and oxidative stress in the context of excess body weight.

## Figures and Tables

**Figure 1 nutrients-17-00163-f001:**
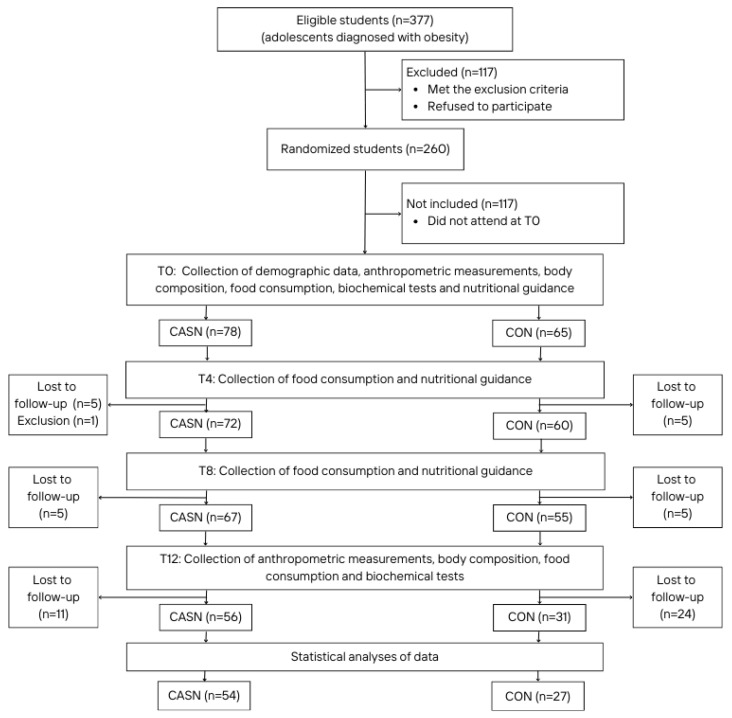
Flowchart of the intervention and data analysis according to the control group (CON) and the cashew nut group (CASN).

**Figure 2 nutrients-17-00163-f002:**
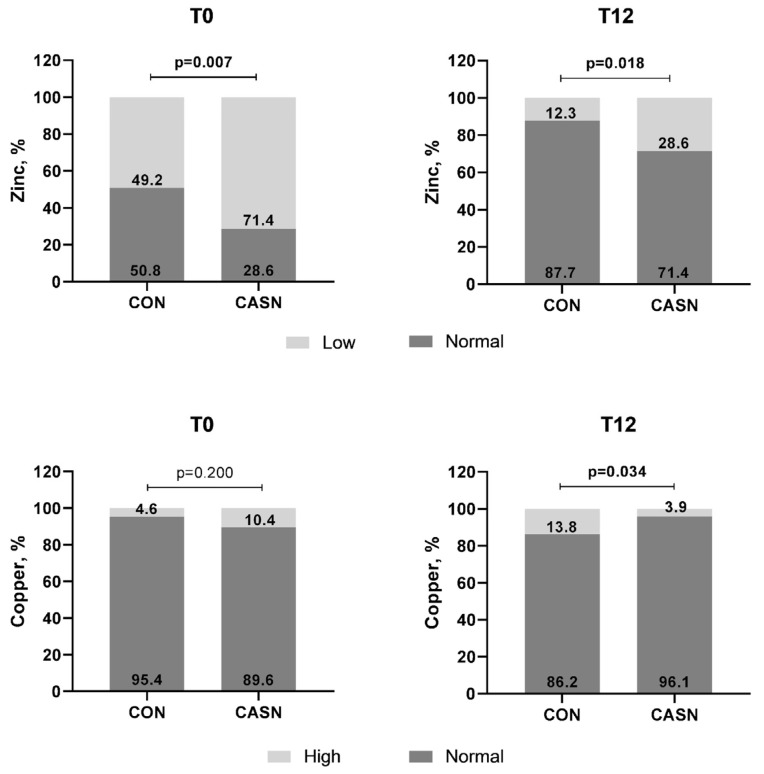
Nutritional status of plasma copper and zinc in the control group (CON) and the cashew nut group (CASN), according to the intervention time.

**Figure 3 nutrients-17-00163-f003:**
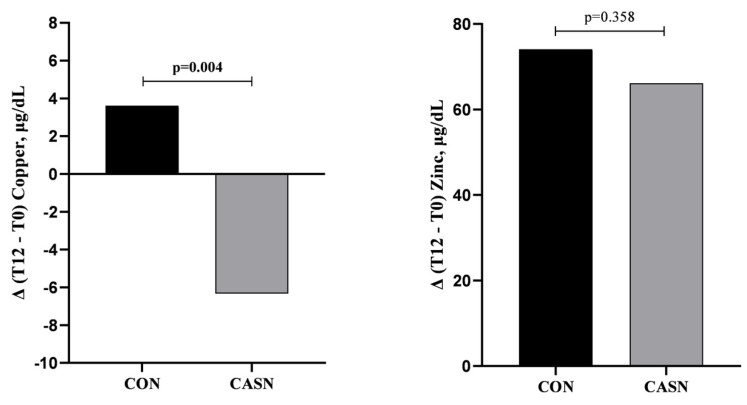
Graph of the variation in biochemical markers after 12 weeks of intervention in the control group (CON) and the cashew nut group (CASN).

**Table 1 nutrients-17-00163-t001:** Nutritional composition of roasted cashew nut kernels (*Anacardium occidentale* L.).

Components	30 g	100 g
Total Carbohydrates, g	9.08	30.27
Protein, g	6.24	20.79
Total Fats, g	14.26	47.54
Copper, mg	0.56	1.87
Zinc, mg	1.32	4.39
Moisture, g	0.39	1.29
Ash, g	0.46	1.54

**Table 2 nutrients-17-00163-t002:** Characterization of the sociodemographic aspects of the study population according to the CON and the CASN.

Variables	All	CON	CASN	*p*-Value
Sex, *n* (%)	*n* = 142	*n* = 65	*n* = 77	0.346
Female	66 (46.5)	33 (50.0)	33 (50.0)	
Male	76 (53.5)	32 (42.1)	44 (57.9)	
Race, *n* (%)	*n* = 138	*n* = 62	*n* = 76	0.801
White	24 (17.6)	11 (49.8)	13 (54.2)	
Black	17 (12.5)	9 (52.9)	8 (47.1)	
Mixed race	95 (69.9)	42 (44.2)	53 (55.8)	
Mother’s Education Level, *n* (%)	*n* = 119	*n* = 52	*n* = 67	0.278
Higher education	12 (10.1)	6 (50.0)	6 (50.0)	
Secondary education	67 (56.3)	25 (37.3)	42 (62.7)	
Primary education/Illiterate	40 (33.6)	21 (52.5)	19 (47.5)	
Type of housing, *n* (%)	*n* = 142	*n* = 65	*n* = 77	0.975
Owned	85 (59.9)	39 (45.9)	46 (54.1)	
Rented	57 (40.1)	26 (45.6)	31 (54.4)	
Paved street, *n* (%)	*n* = 142	*n* = 65	*n* = 77	0.286
Yes	110 (77.5)	53 (48.2)	57 (51.8)	
No	32 (22.5)	12 (37.5)	20 (62.5)	
Basic sanitation, *n* (%)	*n* = 141	*n* = 65	*n* = 76	0.095
Yes	123 (87.2)	60 (48.8)	63 (51.2)	
No	18 (12.8)	5 (27.8)	13 (72.2)	

Abbreviations: CASN, Cashew Nut group; CON, Control group. Differences between the study groups tested with Pearson’s chi-square test. *p*-values < 0.05 were considered significant.

**Table 3 nutrients-17-00163-t003:** Characterization of anthropometric and dietary variables in the CON and the CASN, according to intervention time.

Variables	CON (*n* = 65)			CASN (*n* = 77)			*p*-Value (T0)	*p*-Value (Time and Group)
	T0	T12	*p*-Value	T0	T12	*p*-Value		
Anthropometric								
Weight, kg	74.51 (15.10)	74.76 (15.46)	0.629	74.00 (13.37)	73.62 (13.48)	0.423	0.830	0.370
Height, cm	158.84 (9.54)	160.04 (9.41)	**<0.001**	159.85 (9.87)	161.14 (9.70)	**<0.001**	0.540	0.534
BMI, kg/m^2^ ¥	29.57 (4.17)	28.87 (3.90)	**0.005**	28.78 (3.37)	28.09 (2.97)	**0.002**	0.257	0.979
Lean mass, kg	49.10 (10.18)	48.75 (9.22)	0.518	48.52 (9.72)	48.31 (8.35)	0.738	0.729	0.803
Lean mass, % ¥	70.93 (2.48)	70.64 (2.41)	0.560	71.18 (2.33)	71.15 (2.27)	0.666	0.532	0.471
Fat mass, kg	26.47 (8.93)	26.19 (7.43)	0.644	24.83 (6.40)	25.34 (6.90)	0.351	0.206	0.332
Fat mass, %	34.55 (7.43)	34.41 (5.21)	0.843	33.51 (5.50)	34.01 (4.66)	0.435	0.338	0.500
Dietetics								
Energy, kcal ¥	1788.39 (818.14)	1720.06 (696.49)	0.891	1691.40 (591.35)	1712.77 (811.93)	0.888	0.897	0.995
Protein, g ¥	73.57 (33.81)	68.69 (32.28)	0.491	74.83 (32.66)	70.39 (30.71)	0.260	0.592	0.797
Protein, % ¥	16.99 (6.53)	16.35 (4.83)	0.753	18.10 (6.05)	16.87 (4.37)	0.256	0.202	0.591
Carbohydrate, g ¥	207.04 (95.29)	215.34 (89.62)	0.275	196.86 (78.16)	202.96 (83.21)	0.653	0.927	0.616
Carbohydrate, % ¥	46.6 (9.92)	50.90 (10.40)	**0.010**	46.56 (9.18)	48.66 (9.78)	0.170	0.964	0.322
Lipid, g ¥	62.82 (32.77)	62.46 (27.50)	0.450	65.89 (28.48)	60.82 (23.07)	0.247	0.243	0.181
Lipid, % ¥	32.01 (8.40)	32.03 (5.50)	0.456	34.88 (7.75)	32.88 (8.30)	0.101	**0.029**	0.098
MUFA, g ¥	19.41 (9.99)	19.66 (11.61)	0.968	21.22 (9.83)	20.23 (9.43)	0.450	0.245	0.588
PUFA, g ¥	13.29 (10.49)	12.71 (6.92)	0.626	14.69 (8.56)	13.01 (6.31)	0.168	0.105	0.197
Copper, mg ¥	2.62 (8.51)	1.01 (1.68)	0.082	1.19 (1.83)	0.82 (0.57)	0.323	0.429	0.539
Zinc, mg ¥	9.52 (5.50)	6.86 (2.83)	**0.001**	8.13 (3.53)	7.45 (2.96)	0.264	0.248	0.536

Abbreviations: CASN, cashew nut group; CON, control group; T0, initial time (baseline); T12, final time; BMI, body mass index; MUFA, monounsaturated fatty acids; PUFA, polyunsaturated fatty acids. ¥: transformed variables. *p*-values (within-group, T0 and time and group interaction): Mixed repeated measures ANOVA corrected by Bonferroni post hoc test. *p*-values < 0.05 were considered significant and indicated in bold.

**Table 4 nutrients-17-00163-t004:** Characterization of CON and CASN biochemical variables, according to intervention time.

Variables	CON (*n* = 65)			CASN (*n* = 77)			*p*-Value (T0)	*p*-Value (Time and Group)
	T0	T12	*p*-Value	T0	T12	*p*-Value		
Copper, µg/dL	110.00 (24.21)	113.61 (25.49)	0.156	107.77 (22.78)	101.43 (23.47)	**0.007**	0.573	**0.004**
Zinc, µg/dL ¥	85.82 (46.09)	115.92 (89.27)	**0.038**	67.03 (11.99)	133.20 (75.68)	**0.035**	**0.010**	0.378
SOD, U/gHb *	4258.32 (906.27)	4484.41 (729.58)	0.321	4492.87 (890.46)	4845.69 (907.97)	**0.030**	0.270	0.649
Ratio Cu/Zn ¥	1.49 (0.49)	1.00 (0.66)	**<0.001**	1.63 (0.33)	1.04 (0.59)	**<0.001**	**0.013**	0.504

Abbreviations: CASN, cashew nut group; CON, control group; T0, initial time (baseline); T12, final time; SOD, superoxide dismutase; Cu, copper; Zn, zinc. ¥: transformed variables. *: Data from 27 participants in the CON group and 54 in the CASN group. *p*-values (within-group, T0 and time and group interaction): Mixed repeated measures ANOVA corrected by Bonferroni post hoc test. *p*-values < 0.05 were considered significant and indicated in bold.

## Data Availability

Data will be available upon reasonable request to the corresponding author.

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
