# Peer review of "Effect of Cashew Nut Consumption on Biomarkers of Copper and Zinc Status in Adolescents with Obesity: A Randomized Controlled Trial"

_nutrients, 2024, doi:10.3390/nu17010163_

Round 1

Reviewer 1 Report

Comments and Suggestions for Authors

The manuscript is original, but has several defects.

1) The introduction does not allow the hypothesis to be visualized.

2) If the study is based on chronic inflammation and obesity, the authors should incorporate some marker of inflammation

3) the importance of zinc and copper needs to be explained.

4) There is no comment on how Zinc and copper are absorbed. They do not mention that they compete for absorption.

5) Inflammation parameters must be included in the methodology, if the discussion is based on that point.

7) The reason for dropping out of the study is not included in the results.

8)What is the intra- and inter-sample variation of the measurement of zinc, copper and SOD, to see the importance of the results?

9) How do you explain the lower zinc consumption of the control group during the study?

10) How do you explain the doubling of zinc levels if there was no significant increase in consumption?

11) Was the statistical analysis performed blindly?

In conclusion, the study requires major corrections to be published and new analyzes to answer the author’s question.

Author Response

Reviewer 1

Comments and Suggestions for Authors

The manuscript is original, but has several defects.

Answer: We would like to express our gratitude for the time and effort you dedicated to reviewing our article. We appreciate your constructive comments and suggestions, which were instrumental in improving the quality and clarity of our work. We have provided clarifications in response to the comments and have made revisions based on the suggestions where applicable.

  1. The introduction does not allow the hypothesis to be visualized.

Answer: Thank you for this pertinent comment. We assume that the hypothesis was not clear in the text, so that we have included it in the Introduction section of the revised version of the manuscript as follows:

Introduction:

(lines 86-93): “Due to its nutritional properties, cashew nut has been studied for its ability to control chronic and acute inflammatory and oxidative processes. However, there is a gap in the scientific literature regarding the effects of cashew nut consumption on the Cu and Zn status. Based on promising results from experimental research [26-29], we hypothesized that their consumption in combination with a healthy diet could produce similar results in a clinical setting, which in turn might contribute to the control and prevention of chronic diseases [30,31]. However, evidence concerning the effects of cashew nut consumption on the concentration of these essential minerals are still limited.”

  1. If the study is based on chronic inflammation and obesity, the authors should incorporate some marker of inflammation.

Answer: We appreciate the reviewers’ suggestion to include inflammatory markers in the study, which would allow us to examine the role of inflammatory response on the association between cashew nut consumption and plasma concentrations of copper and zinc. However, data on inflammatory markers were not available, which prevented us from including them in the analyses. In addition, the present study focused on copper and zinc status in response to cashew nut consumption, and chronic inflammation, oxidative stress and obesity were mentioned as potential biological mechanisms underlying the associations found. We have rephrased the study conclusions highlighting that these mechanisms should be investigated in future studies. The rephrase conclusions reads as follows:

(lines 438-443): “The consumption of cashew nuts reduced plasma Cu levels in adolescents with obesity. On the other hand, nutritional guidance activities may influence increased plasma Zn levels and reduced Cu/Zn ratio in adolescents with obesity. Notably, similar results were observed in both the CASN and CON groups with respect to body composition and Cu and Zn intake. Therefore, further studies are needed to investigate the potential mechanisms underlying changes in Cu and Zn levels following cashew nut consumption, with particular focus on their roles in regulating inflammation and oxidative stress in the context of excess body weight.”

  1. The importance of zinc and copper needs to be explained.

Answer: The reviewer is correct, and we agree that the central role of zinc and copper in our study deserves further elaboration. In response, we have expanded the rationale regarding the biological role of these micronutrients, and their relevance to the context of the study.  The new version can be found in the revised manuscript, lines 62-67 and 70-75, as follows:

Introduction:

(lines 62-67): “These micronutrients are essential and have critical functions in biological processes. Cu plays a catalytic role in cuproenzymes, performing redox cycles necessary for cellular respiration, free radical detoxification, and neurotransmitter biosynthesis [16]. Zn is a component of more than 300 enzymes [10] and plays important roles in transcription factors, DNA-binding proteins, and cellular regulation through kinases, phosphatases, and transport channels [17].”

(lines 69-74): “The enzyme superoxide dismutase (SOD), which relies on these minerals, facilitates the conversion of superoxide radicals into hydrogen peroxide and helps maintain cellular homeostasis [18]. Both minerals also compete for intestinal absorption, and this competition can influence the availability of minerals in the body, especially in situations of unbalanced intake or deficiency [19].”

  1. There is no comment on how Zinc and copper are absorbed. They do not mention that they compete for absorption.

Answer: We acknowledge the importance of reporting the interaction between the two micronutrients. We have added this information to the introduction as recommended. Please find the revised version in lines 72-74 and as follows:

Introduction:

(lines 72-74): “Both minerals also compete for intestinal absorption, and this competition can influence the availability of minerals in the body, especially in situations of unbalanced intake or deficiency [19].”

  1. Inflammation parameters must be included in the methodology, if the discussion is based on that point.

Answer: We appreciate your insightful comment regarding the inclusion of inflammatory markers. As noted in our response to point 2, data on these parameters were not available and could not be included in the analyses. Our study focused primarily on copper and zinc status in response to cashew nut consumption, with chronic inflammation, oxidative stress, and obesity discussed as potential mechanisms underlying the observed associations. We have clarified this in the revised text to ensure greater precision and understanding.

  1. The reason for dropping out of the study is not included in the results.

Answer: Thank you for your comment. We recognize the importance of reporting the reasons for dropout. We considered these dropouts to be random because they occurred without the influence of specific factors. Many adolescents dropped out of the data collection by their own choice, primarily due to fear of having their blood samples collected. The absence of adolescents from school on the day of data collection was another factor that influenced the dropout rate. This information has been added to the revised manuscript. Please find it in the Methods section and as follows:

Methods

(lines 274-276): “The number of dropouts was influenced by various factors, including the absence of adolescents from school on the day of data collection, and the fear of having blood samples taken, as reported by some adolescents.”

  1. What is the intra- and inter-sample variation of the measurement of zinc, copper and SOD, to see the importance of the results?

Answer: The intra- and inter-group analyses are complementary and provide a broader understanding of our results regarding Zn, Cu, and SOD measurements. The intra-group analysis allowed us to identify significant changes over time, which is particularly useful for evaluating the influence of the intervention within each group. Given the approach of nutritional guidance combined with cashew nut consumption used in the present study, we considered this analysis particularly important to show how each group individually responded to the intervention.

However, the intergroup analyses are the most critical for confirming the effectiveness of cashew nut consumption, as they indicate whether the effect in the intervention group (CASN) was different from that in the CON group. Specifically, the "time by group interaction" analysis revealed a reduction in Cu levels in the CASN group while controlling for confounding effects due to the duration of the study or other external factors. Additionally, we conducted a delta (Δ) analysis to quantify the magnitude of the changes that could be attributed to the intervention.

We believe that the intra- and intergroup variation measurements offer distinct perspectives on our results. Together, they enhance the robustness of the conclusions, in addition to providing a more detailed and reliable view of the effects of the cashew nut intervention on Zn, Cu, and SOD levels.

  1. How do you explain the lower zinc consumption of the control group during the study?

Answer: We believe that the nutritional guidance provided during the study encouraged changes in the participants' food choices, potentially leading to a reduction in the consumption of certain ultra-processed foods, which are often fortified with micronutrients, including zinc. This finding was not observed in the CASN group, likely because cashew nuts are a natural source of zinc, which may have compensated for any changes in dietary intake. We have added an explanation for this in lines 380-383, as follows:

“The CON group presented a slight reduction in dietary Zn intake, possibly due to changes in food choices following nutritional guidance such as reducing the consumption of certain ultra-processed foods, which are often fortified with micronutrients, including zinc. On the other hand, both the CON and CASN groups presented relatively high plasma Zn concentrations and reduced Cu/Zn ratios after 12 weeks.”

  1. How do you explain the doubling of zinc levels if there was no significant increase in consumption?

Answer: We suggest that the doubling of Zn levels results from the modulation of inflammatory factors. Zn is primarily transported by albumin and tends to decrease in response to inflammation. Thus, the improvement in the inflammatory state may be directly related to the increase in Zn levels, without the need for increased micronutrient intake. This explanation is presented throughout the discussion in lines 389-395:

“Cu and Zn concentrations can be modulated in response to inflammation in the body [56]. Cu is positively regulated in the bloodstream as a positive acute-phase reactant and is bound mainly to ceruloplasmin. In contrast, Zn is mainly transported by albumin and tends to decrease in response to inflammation [57]. This inverse behavior with respect to inflammatory cytokines may contribute to antagonistic changes in trace elements [10]. Thus, improving the inflammatory state may be directly related to a decrease in Cu and a concomitant increase in Zn.”

  1. Was the statistical analysis performed blindly?

Answer: The statistical analysis was not blinded. However, to mitigate potential bias, we relied on objective data collection procedures, followed a pre-specified analysis plan, performed all statistical analyses using rigorous methods to control for confounding variables and adhered to transparent reporting practices to ensure the robustness and reliability of our findings.

  1. In conclusion, the study requires major corrections to be published and new analyzes to answer the author’s question.

Answer: We appreciate the suggestion and recognize the importance of the proposed changes. As a result, we have revised the conclusion to enhance clarity and understanding. In the hope that we have addressed all the reviewer’s concerns, we remain open to additional comments and suggestions.

Reviewer 2 Report

Comments and Suggestions for Authors

Scientists decided to present studies based on evaluation of influence of cashew nut consumption on biomarkers of copper and zinc in adolescents with obesity. The issue isvery important due to the fact that obesity is a huge problem, especially in adolescents. It is known that mealions have big impact on our health and disturbance of their level impact on various diseases such as obesity, cardiovascular problems or neurodegeneration. The presented studies are interesting and well prepared. The paper is suitable for publication in the Journal but some points have to be improved:

Abstract: informative and readable

Introduction: based on up to date references, includes all necessary information

Methods: Were there any challenges or limitations in using multiple food composition tables, particularly in terms of variability in nutrient data between sources?

Were participants' hydration levels assessed or controlled, as hydration status can significantly affect bioelectrical impedance results?

How was compliance with the fasting, exercise, and smoking restrictions monitored or ensured for the bioelectrical impedance analysis?

Were there any significant findings linking sociodemographic factors to dietary intake, anthropometric data, or biochemical markers?

Results: presented in for of charts and detail comments. This section is well prepared.

Discussion: detail, based on up to date references .

Conclusions: please provide more details, the present form is insufficient

References: adequate

Author Response

Reviewer 2

Comments and Suggestions for Authors

Scientists decided to present studies based on evaluation of influence of cashew nut consumption on biomarkers of copper and zinc in adolescents with obesity. The issue is very important due to the fact that obesity is a huge problem, especially in adolescents. It is known that mealions have big impact on our health and disturbance of their level impact on various diseases such as obesity, cardiovascular problems or neurodegeneration. The presented studies are interesting and well prepared.

Answer: We would like to express our gratitude for the time and effort you dedicated to reviewing our article. We appreciate your constructive comments and suggestions, which were instrumental in improving the quality and clarity of our work. We have provided clarifications in response to the comments and have made revisions based on the suggestions where applicable. Furthermore, we conducted a comprehensive review of the manuscript and corrected any writing or formatting errors, which are now highlighted. These revisions include the correction of the number in Figure 1, the addition of ethylenediamine tetraacetic anticoagulant to the biochemical markers, the substitution of the term "obese" with "with obesity", the addition of a comma in Table 1, and the inclusion of the term "p-value (T0)" in Table 3.

 The paper is suitable for publication in the Journal but some points have to be improved:

  1. Abstract: informative and readable

Answer: Thank you.

  1. Introduction: based on up to date references, includes all necessary information

Answer: We appreciate your feedback. In response to suggestions from another reviewer, we have added some information to the introduction. We hope that these changes have contributed to greater clarity in conveying the ideas presented.

  1. Methods: Were there any challenges or limitations in using multiple food composition tables, particularly in terms of variability in nutrient data between sources?

Answer: The prioritization of Brazilian food composition tables, such as the Brazilian Food Composition Table and the Pesquisa de Orçamento Familiar Table, aimed to ensure that the nutritional data more accurately reflected local dietary habits and the composition of foods consumed by the study participants. The Brazilian Food Composition Table, which includes more than 5,700 food items, including more than 4,000 dietary adaptations for different dietary contexts (e.g., gluten-free, lactose-free, vegan, and vegetarian foods), was used as the primary source because of its comprehensiveness and detail. However, the need to use multiple tables arose because certain regional foods specific to the study area were not included in this primary table. As a secondary source, the Pesquisa de Orçamento Familiar Table, developed from household surveys conducted across all regions of Brazil, was included, providing additional data on typical foods. Only in isolated cases, where no Brazilian table covered the required foods, was the USDA database used as a last resort. This approach ensured the completeness of the nutritional analysis while prioritizing the cultural and dietary representativeness of the data used.

  1. Were participants' hydration levels assessed or controlled, as hydration status can significantly affect bioelectrical impedance results?

Answer: Participants were instructed to fast for 8-12 hours and abstain from water and food, to minimize the influence of hydration status on bioelectrical impedance results. This information was added on line 194.

  1. How was compliance with the fasting, exercise, and smoking restrictions monitored or ensured for bioelectrical impedance analysis?

Answer: Adherence to the fasting, exercise, and smoking restrictions was monitored by self-reporting by the adolescents and their respective parents or guardians. Both were thoroughly informed before and during the study of the importance of adherence to the protocol to ensure accurate results. To further promote compliance, assessments were scheduled in the early morning hours, approximately 7:00 AM, to minimize the likelihood of deviations from the protocol.

  1. Were there any significant findings linking sociodemographic factors to dietary intake, anthropometric data, or biochemical markers?

Answer: In this study, sociodemographic data were assessed as covariates to ensure the homogeneity of the groups. The analysis confirmed that the groups were indeed homogeneous. Furthermore, we carefully recruited participants from both groups within the same school and during the same time, minimizing potential biases related to temporal or environmental factors. Given these measures, we are confident that sociodemographic factors do not significantly influence the study results, thus ensuring the reliability of our findings.

  1. Results: presented in for of charts and detail comments. This section is well prepared.

Answer: We appreciate the positive feedback from the reviewer.

  1. Discussion: detail, based on up to date references.

Answer: Thank you for recognizing the depth of our discussion.

  1. Conclusions: please provide more details, the present form is insufficient

Answer: We appreciate your suggestion. In response, we have expanded the conclusion to provide more comprehensive insights and further clarify the implications of our findings, as well as the need of further studies on the mechanisms underlying the results found. The revised version can be found in lines 438-443.

  1. References: adequate

Answer: Thank you for your assessment. We hope we have addressed all the reviewer's concerns and welcome any additional comments or suggestions.